# The effectiveness of acellular nerve allografts compared to autografts in animal models: A systematic review and meta-analysis

Berend O. Broeren[1]*, Caroline A. Hundepool[2], Ali H. Kumas[1], Liron S. Duraku[3], Erik T. Walbeehm[4], Carlijn R. Hooijmans[5,6], Dominic M. Power[7], J. Michiel Zuidam[2], Tim De Jong[1]

1 Department of Plastic & Reconstructive Surgery, Radboud University Medical Centre, Nijmegen, The Netherlands, 2 Department of Plastic & Reconstructive Surgery, Erasmus MC, Rotterdam, The Netherlands, 3 Department of Plastic, Reconstructive & Hand Surgery, Amsterdam UMC, Amsterdam, The Netherlands, 4 Department of Plastic, Reconstructive & Hand Surgery, Haga Hospital and Xpert Clinic, Den Haag, The Netherlands, 5 Department for Health Evidence Unit SYRCLE, Radboud University Medical Centre, Nijmegen, The Netherlands, 6 Department of Anesthesiology, Pain and Palliative Care (Meta Research Team), Radboud University Medical Centre, Nijmegen, The Netherlands, 7 Department of Hand & Peripheral Nerve Surgery, Queen Elizabeth Hospital, Birmingham, United Kingdom

* berend.broeren@radboudumc.nl

## Abstract

### Background

Treatment of nerve injuries proves to be a worldwide clinical challenge. Acellular nerve allografts are suggested to be a promising alternative for bridging a nerve gap to the current gold standard, an autologous nerve graft.

### Objective

To systematically review the efficacy of the acellular nerve allograft, its difference from the gold standard (the nerve autograft) and to discuss its possible indications.

### Material and methods

PubMed, Embase and Web of Science were systematically searched until the 4th of January 2022. Original peer reviewed paper that presented 1) distinctive data; 2) a clear comparison between not immunologically processed acellular allografts and autologous nerve transfers; 3) was performed in laboratory animals of all species and sex. Meta analyses and subgroup analyses (for graft length and species) were conducted for muscle weight, sciatic function index, ankle angle, nerve conduction velocity, axon count diameter, tetanic contraction and amplitude using a Random effects model. Subgroup analyses were conducted on graft length and species.

### Results

Fifty articles were included in this review and all were included in the meta-analyses. An acellular allograft resulted in a significantly lower muscle weight, sciatic function index, ankle

**Data Availability Statement:** All relevant data are within the paper and its supporting information files.

**Funding:** The authors will receive an award from ZonMw upon publication. Partially to make open access publication possible. ZonMw is an independent institute which has no benefits from the publicated data of this article. Therefore, the funders had no role in study design, data collection and analysis, decision to publish, or preparation of the manuscript. Initials receiving author: T. De Jong Grant number: 114024160 Name funder: ZonMw URL:https://www.zonmw.nl/nl/ The funders had no role in study design, data collection and analysis, decision to publish, or preparation of the manuscript.

**Competing interests:** Conflict of Interest Liron S. Duraku and Dominic M. Power: Both authors are on the educational committee of the Global Nerve Foundation which is a non-profit organization of which Axogen is a founding Member. Both authors do not get any financial compensation for their role in the Global Nerve Foundation. This does not alter our adherence to PLOS ONE policies on sharing data and materials.

angle, nerve conduction velocity, axon count and smaller diameter, tetanic contraction compared to an autologous nerve graft. No difference was found in amplitude between acellular allografts and autologous nerve transfers. Post hoc subgroup analyses of graft length showed a significant reduced muscle weight in long grafts versus small and medium length grafts. All included studies showed a large variance in methodological design.

## Conclusion

Our review shows that the included studies, investigating the use of acellular allografts, showed a large variance in methodological design and are as a consequence difficult to compare. Nevertheless, our results indicate that treating a nerve gap with an allograft results in an inferior nerve recovery compared to an autograft in seven out of eight outcomes assessed in experimental animals. In addition, based on our preliminary post hoc subgroup analyses we suggest that when an allograft is being used an allograft in short and medium (0-1cm, > 1-2cm) nerve gaps is preferred over an allograft in long (> 2cm) nerve gaps.

## Introduction

Peripheral nerve injuries affect 2,8% of all trauma cases, and despite surgical repair, they often result in deterioration of quality of life for these patients [1–3]. In some injuries, a segmental loss of a peripheral nerve occurs after trauma or tumor excision for example [4, 5].

The current gold standard for the surgical repair of a peripheral nerve injury that cannot be directly coaptated is a nerve autograft. The sural nerve is the most commonly used because it supplies a consistent source of graft material and is anatomically accessible [6]. However, this procedure has several limitations. The length of donor nerve available and its limited diameter are often insufficient to achieve a complete reconstruction of multiple or significant segmental defects. Besides, the procedure may cause considerable donor site morbidity, such as pain and loss of sensation [7–9]. A specific form of autograft, a vascularized nerve graft, resulted in a superior nerve recovery. However, it could result in an even more considerable donor site morbidity [10].

Several techniques have been investigated to replace the nerve autograft, including allografts, biological conduits and synthetic conduits [11–13]. All with their benefits and drawbacks. Of these options, the acellular allograft seems the most promising [14]. These grafts provide the needed internal structural and molecular composition of the extracellular matrix in a cell-free scaffold, which supports nerve regeneration while retaining a nonimmunogenic nature [15]. This procedure however, has some drawbacks as well, including uncertain histocompatibility and ethical and legal concerns.

A variety of methods have been studied to prepare an acellular allograft such as cold preservation, freeze-thaw cycling, chemical detergent and enzymatic preparations, lyophilization and irradiation [16–18]. There is only one such method that is FDA approved available to surgeons produced by AxoGen, Inc., Alachua, Florida. AxoGen develops its human allografts by combining proprietary detergent processing and gamma irradiation which removes cellular remnants while minimizing the microstructural damage. Next to that, chondroitin sulfate proteoglycan is enzymatic removed from the endoneural tube system to advance axon regeneration [19]. Several additions and alterations to this method have been researched. However, at this moment these techniques are not clinically available.

There is little clinical and experimental evidence about the difference in outcomes between an acellular allograft and an autograft. Therefore, A systematic review of experimental studies was conducted to investigate the efficacy of the acellular nerve allograft, its difference from the gold standard (the nerve autograft) and to discuss the possible indications for the use of an allograft.

## Material and methods

### Research protocol

Before starting this systematic review, a protocol was defined in advance and registered in an international database (PROSPERO, registration number CRD42020186451). The PRISMA guidelines for conducting a systematic review were followed.

### Search strategy

PubMed (Medline), Embase (OVID) and Web of Science were systematically searched to identify all original articles. The search contained studies up to the 4th of January 2022. Search terms included 'nerve reconstruct', 'nerve transfer', 'nerve graft', 'allograft', 'allogeneic', 'acellular', 'decellularize' and their synonyms in abstract and title fields (see S1 Table for the complete search strategy). To identify all animal studies, the SYRCLE search filters were used [20, 21]. Endnote (Clarivate Analytics, Pennsylvania, USA) was used to remove duplicates. Two authors (BOB and AK) performed the screening process independently using Rayyan web tool [22]. All titles and abstracts were screened to determine their relevance by utilizing the pre-established inclusion and exclusion criteria. Reference lists of the remaining studies were screened manually for potentially relevant new studies. Full text screening of all relevant articles was done by two reviewers for final selection. Divergences were solved by consensus discussion. Any remaining divergences were solved by consulting TDJ as a third reviewer.

### Inclusion and exclusion criteria

We included an original peer reviewed paper 1) that presented distinctive data; 2) made a clear comparison between not immunologically processed acellular allografts and autologous nerve transfers; 3) was performed in laboratory animals of all species and sex; 4) investigated the effect of acellular allografts on motor outcomes: sciatic function index, muscle weight (gram), ankle angle (degrees), electrophysiology (nerve conduction velocity (ms/s), amplitude (mA) or latency (ms)) and sensory outcomes: hot-cold testing, pin-prick testing, Semmes-Weinstein testing and histomorphometry (axon count and diameter). No publication date restriction was applied.

### Critical appraisal

Two authors (BOB and AK) independently assessed the risk of bias using the SYRCLE's tool for assessing the risk of bias for animal studies. This appraisal was subsequently merged by consensus and disagreements were solved by discussion [23]. A "yes" indicating a low risk of bias, a "no" indicating a high risk of bias or a "?" indicating an unknown risk of bias was scored for all criteria. We determined selective outcome reporting by establishing if all outcome measures stated in the material and methods section were also reported in the results. Baseline characteristics were: species, age and weight. We included two items to overcome the problem of judging to many items as "unclear risk of bias: reporting on any measure of randomization and reporting on any measure of blinding. For these two questions a "yes" indicates reported and a "no" indicates not reported.

## Data extraction

From the included studies, both reviewers (BOB and AK) extracted the data in duplicate. The descriptive data included: first author's name, the year of publication, studied species, sex, total number of animals, number of grafts, studied nerve, studied muscle, graft size and time points. The mean, standard deviation (SD) and total number of subjects (n) were recorded for all outcomes. In case multiple locations per nerve were reported, we used the most distal segment of the graft. When the SEM was reported, it was recalculated to SD (SD = SEM x $\sqrt{n}$). If data were only presented graphically, Universal Desktop Ruler software (https://avpsoft.com/products/udruler/) was used by two reviewers independently to measure a fair estimation of the presented data, after that the mean of these two independent measurements was used. We attempted to contact the authors for additional information in case relevant data were missing.

## Statistical analysis

Comprehensive Meta-Analysis (CMA version 3.3) was used to analyze all data. The standardized mean difference (SMD) and 95% confidence interval (95% CL) for all outcome measurements comparing acellular allografts and conventional autografts were calculated with Hedges' g correction. A random effects model was applied, which takes the accuracy of independent studies and the variation among studies into account and weighs all studies accordingly. $I^2$ was used to asses heterogeneity. In case a study reported results at different time points using the same experimental group, these results were pooled to obtain an overall SMD with Hedges'g correction using a random-effects model and variance. Subgroup analyses were conducted post hoc for species (rat, rabbit, monkey and dog) and graft lengths (0–1 cm, > 1–2 cm and > 2 cm). We only interpreted the results of subgroup analysis when groups consisted of 5 or more individual studies.

To detect publication bias funnel plots were created and evaluated on symmetry using Egger's regression and Trim and Fill analysis, if there were at least 15 or more independent studies per outcome. We plotted the SMD against a sample size-based precision estimate($1/\sqrt{n}$), because SMDs may cause funnel plot distortion.

A sensitivity analysis was performed to assess the robustness of our findings. The impact of excluding studies published before 2008 and studies that used animals as their own control was evaluated.

# Results

## Study selection process

The systematic literature search presented in S1 Table yielded 1191 unique references (Fig 1 shows a consort flow chart). After title abstract screening, 136 studies met the selection criteria. Finally, after studying the full-text articles, 50 studies were included in the review and meta-analyses.

## Study quality and risk of bias

The general results of our risk of bias assessment of the included references are shown in Fig 2. Poor reporting of essential methodological details in most animal experiments resulted in an unclear risk of bias in most studies. In particular reporting about any randomization and blinding measures taken at any level was 64% (32 out of 50 publications). Assessment of the risk of bias was done separately for the 3 studies that used animals as their own control because some aspects were not applicable (Fig 3).

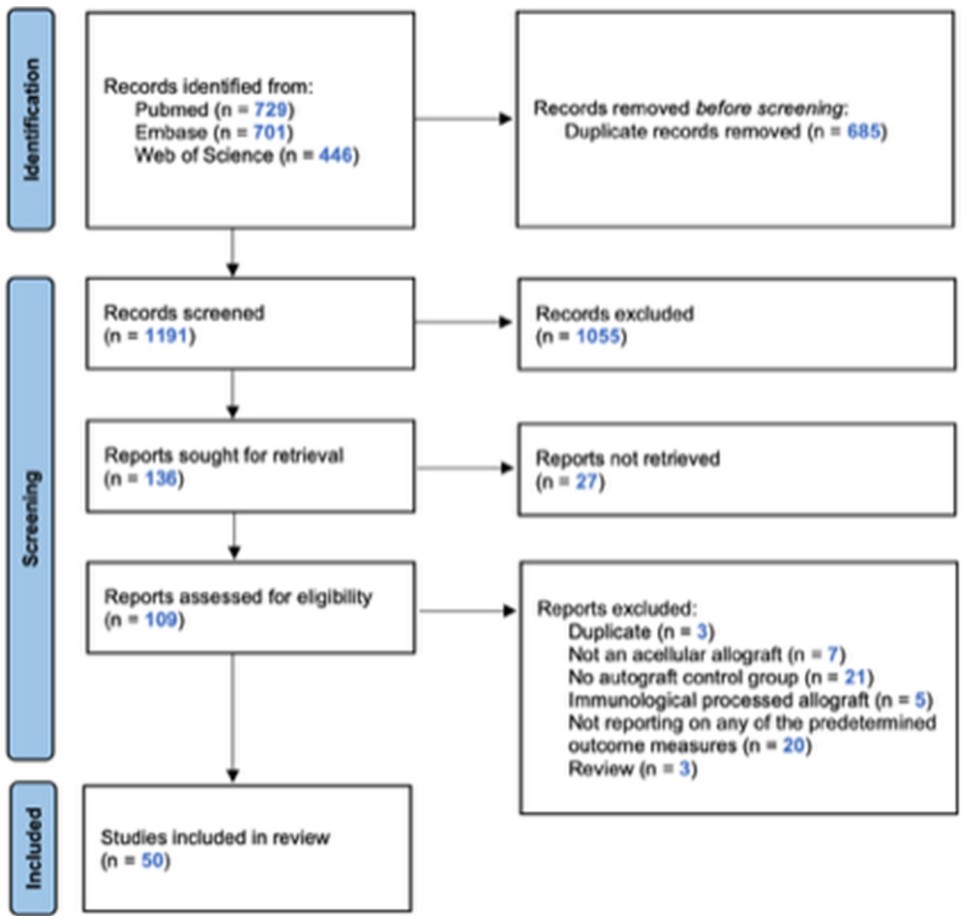

**Fig 1. Flow chart of the study selection.**

## Study characteristics

A summary of the characteristics of the 50 included publications is shown in Table 1 [24–73]. The characteristics per outcome measurement are depicted in the Appendix. The most commonly used specie was rat (80%), followed by rabbit (6%), monkey (6%), dog (6%) and mice (2%). Gender was not reported in 20% (10 of 50) of the publications. Out of the remaining studies 33 used males, 2 used females and in 5 both sexes were used. Different nerves were used with the sciatic nerve being the most common (80%), followed by the peroneal nerve (6%), facial nerve (4%), radial nerve (4%), ulnar nerve (2%), tibial nerve (2%) and femoral nerve (2%).

## Overall analysis

Overall analysis showed a significant lower muscle weight, sciatic function index, tetanic contraction, nerve conduction velocity and smaller ankle angle, axon count diameter after treatment with an acellular allograft compared to an autograft (Shown in Table 2). No significant difference in amplitude was found.

## Subgroup analyses

Subgroup analysis of all subgroups containing a minimum of 5 comparisons revealed a significant difference in muscle weight when comparing graft length between acellular allografts and

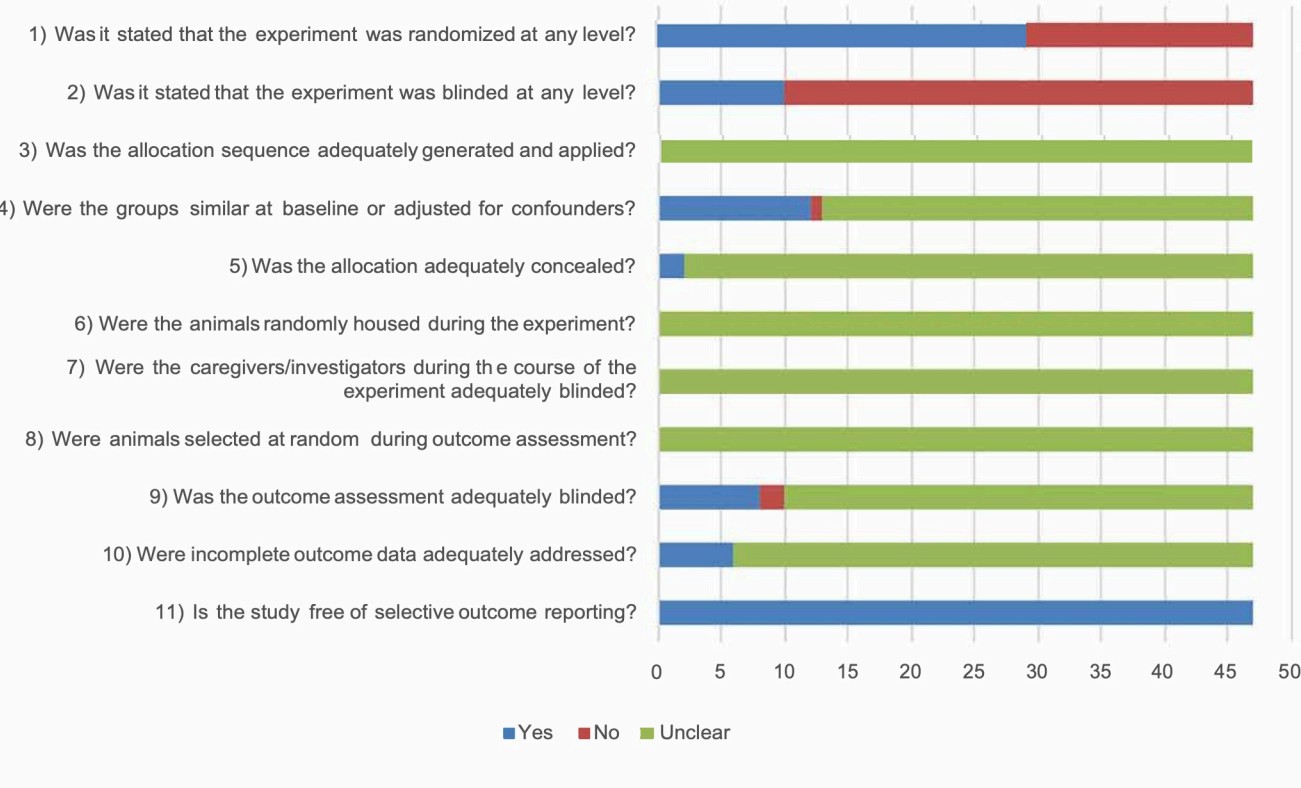

The first two items assess study quality by scoring reporting, a "yes" score indicates reported and a "no" score indicates unreported. The other items assessed risk of bias, with "yes" indicating low risk of bias, "no" high risk of bias, and "?" unclear risk of bias.

**Fig 2. Results of the risk of bias assessment of 47 included studies in this systematic review.** The first two items assess study quality by scoring reporting, a "yes" score indicates reported and a "no" score indicates unreported. The other items assessed risk of bias, with "yes" indicating low risk of bias, "no" high risk of bias, and "?" unclear risk of bias.

conventional nerve autografts. Autografts showed a more favorable result in long grafts (> 2 cm) than in medium and short grafts (0–1 cm, > 1–2 cm) compared to acellular allografts (see Table 3).

However, for nerve conduction velocity and axon count no significant difference was found comparing graft length between acellular allografts and conventional nerve autografts.

All other subgroup analyses on graft length could not be interpreted because groups consisted of fewer than 5 studies. The same goes for all subgroup analyses for species.

## Sensitivity analysis

Exclusion of the studies published before 2008 did not alter our results significantly (see S2 Table). Also, when the studies were excluded in which animals were their own control no significant changes were found, only the amplitude SMD improved significantly (0.70 to 1.00), in favor of autografts (S3 Table).

Conclusions of all subgroup analyses appeared to be robust.

## Publication bias analysis

Publication bias could only be assessed for axonal count, muscle weight and nerve conduction velocity, because all other outcome measurements consisted of fewer than 15 independent studies. The funnel plot for muscle weight and axon count suggested some asymmetry. Duval

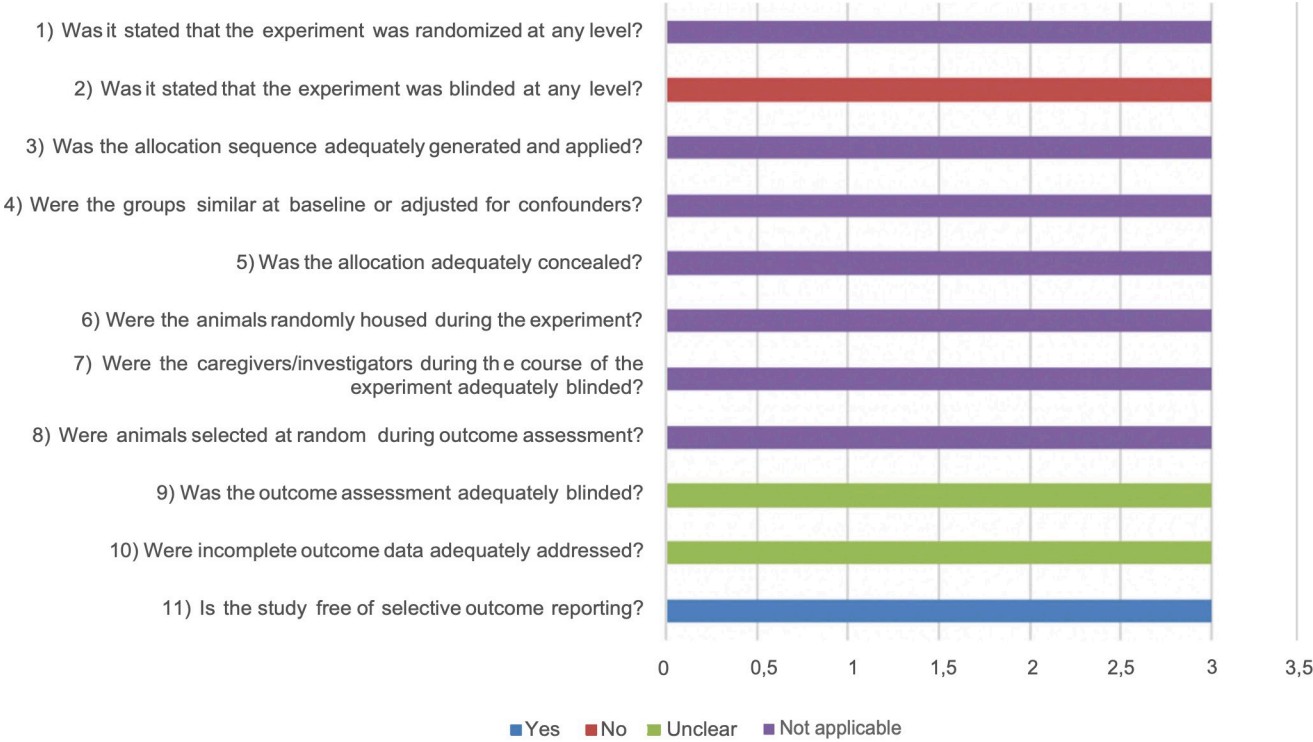

The first two items assess study quality by scoring reporting, a "yes" score indicates reported and a "no" score indicates unreported. The other items assessed risk of bias, with "yes" indicating low risk of bias, "no" high risk of bias, and "?" unclear risk of bias.

**Fig 3. Results of the risk of bias assessment of the 3 included studies in this systematic review where animals were their own control group.** The first two items assess study quality by scoring reporting, a "yes" score indicates reported and a "no" score indicates unreported. The other items assessed risk of bias, with "yes" indicating low risk of bias, "no" high risk of bias, and "?" unclear risk of bias.

and Tweedie's Trim and Fill analysis resulted in 14 and 6 extra data points (Figs 4 and 5), indicating the presence of publication bias and some overestimation of the identified summary effect size. No publication bias for nerve conduction velocity was found (Fig 6).

## Discussion

Our results indicate that treating a nerve gap with an allograft results in an inferior nerve recovery compared to an autograft in seven out of eight outcome measurements assessed in animal models. Our subgroup analysis suggests that when an allograft is being used, an allograft in short and medium (0-1cm, > 1-2cm) nerve gaps performs better than an allograft in long (> 2cm) nerve gaps.

Comparing the available literature regarding the use of acellular allografts was challenging because a large variation in decellularization techniques were used. A variety of methods have been studied to prepare an acellular allograft by these labs such as cold preservation, freeze-thaw cycling, chemical detergent and enzymatic preparations, lyophilization and irradiation [16–18]. There is little to no evidence for what combination of these decellularization methods, give the best nerve regeneration. Fig 7 shows an approximate overview of the different aspects of these methods. We tried to investigate which method led to the best nerve recovery by using the data available in the current literature. Due to the great variation in methods used, groups became too small to perform statistical analysis.

**Table 1. The characteristics of all 50 included references.**

| Reference | Species | Animals (grafts) | Nerve | Graft size (mm) | Time points (weeks) | Outcome measurements | Muscle |
|---|---|---|---|---|---|---|---|
| Hadlock et al., 2001 | Rat | 35 | Sciatic | 7 | 6, 10.5 | SFI | |
| Boriani et al., 2019 | Rabbit | 15 | Tibial | 20 | 12 | NCV<br>Amplitude | |
| Cai et al., 2017 | Rat | 18 | Sciatic | 15 | 8 | Muscle weight<br>Diameter<br>Axonal count | Gastrocnemicus |
| Chato-Astrain et al., 2020 | Rat | 24 | Sciatic | 10 | 12 | SFI<br>Muscle weight | Gastrocnemicus |
| Dai et al., 2014 | Rat | 50 | Sciatic | 10 | 4, 8, 12 | SFI<br>Diameter<br>Axonal count | |
| Frerichs et al., 2002 | Rat | 40 | Sciatic | 20 | 6 | Axonal count | |
| Gao et al., 2014 | Rat | 45 | Sciatic | 10 | 6, 12 | SFI<br>Muscle weight<br>NCV | Gastrocnemicus |
| Giusti, Lee et al., 2016 | Rat | 88 | Sciatic | 10 | 16 | Muscle weight<br>Ankle angle<br>Axon count<br>Tetanic contraction | Tibialis anterior |
| Giusti, Willems et al., 2012 | Rat | 65 | Sciatic | 10 | 12, 16 | Muscle weight<br>Ankle angle<br>Tetanic contraction | Tibialis anterior |
| Gulati et al., 1990 | Rat | 24 (48) | Sciatic | 20 | 12 | Muscle weight | Extensor digitorum longus |
| Haase et al., 2003 | Rat | 19 | Peroneal | 20, 40 | 3, 15 | Muscle weight<br>Tetanic contraction | Extensor digitorum longus |
| Huang et al., 2015 | Rat | 18 | Facial | 10 | 12 | NCV<br>Diameter<br>Axon count | |
| Hu, Zhang et al., 2010 | Rabbit | 72 | Facial<br>Peroneal | 50 | 4, 12, 24 | Axonal count | |
| Hu, Zhu et al., 2007 | Monkey | 12 | Ulnar | 40 | 24 | NCV | |
| Hundepool et al., 2018 | Rat | 66 | Sciatic | 10 | 12, 16 | Muscle weight<br>Ankle angle<br>Axon count<br>Tetanic contraction | Tibial |
| Ide et al., 1998 | Dog | 16 | Sciatic | 50 | 12 | Axon count | |
| Jiang et al., 2016 | Rat | 40 | Sciatic | 15 | 12 | Muscle weight<br>NCV | Triceps surae |
| Li, Zhao et al., 2013 [41] | Rabbit | 51 | Sciatic | 30 | 24 | NCV<br>Amplitude | |
| Li, Peng et al., 2008 | Rat | 30 | Sciatic | 10 | 4, 8, 12, 16 | SFI<br>Muscle weight<br>Diameter<br>Axon count | Gastrocnemicus<br>Soleus |
| Moore et al., 2011 | Rat | 60 | Sciatic | 14 | 6, 16 | Muscle weight<br>Tetanic contraction | Extensor digitorum longus |
| Muheremu et al., 2016 | Rat | 45 | Femoral | 5 | 12 | Muscle weight<br>Amplitude<br>Diameter | Quadriceps femoris |
| Nakamoto et al., 2021 | Rat | 40 | Sciatic | 5 | 3, 6, 12, 14 | SFI<br>Diameter<br>Axon count | |
| Piao et al., 2020 | Rat | 51 | Sciatic | 20 | 24 | NCV<br>Amplitude | |

*(Continued)*

**Table 1.** (Continued)

| Reference | Species | Animals (grafts) | Nerve | Graft size (mm) | Time points (weeks) | Outcome measurements | Muscle |
|---|---|---|---|---|---|---|---|
| Qiu et al., 2020 | Dog | 15 | Sciatic | 50 | 24 | NCV<br>Diameter<br>Axon count | |
| Rovak et al., 2004 | Rat | 16 | Peroneal | 20 | 15 | Muscle weight<br>Axon count | Extensor digitorum longus |
| Saheb et al., 2013 | Rat | 30 | Sciatic | 20, 40, 60 | 10, 20 | Muscle weight | Extensor digitorum longus |
| Shin et al., 2019 | Rat | 20 | Sciatic | 10 | 4, 8, 12, 16 | Muscle weight<br>Ankle angle<br>Axon count<br>Tetanic contraction | Tibialis anterior |
| Sun et al., 2009 | Rat | 24 | Sciatic | 10 | 12 | NCV<br>Amplitude | |
| Tang, Kilic et al., 2013 | Rat | 54 | Sciatic | 10 | 6, 12 | Muscle weight<br>Tetanic contraction | Tibialis anterior |
| Tang, Whiteman et al., 2019 | Rat | 81 | Sciatic | 10 | 12, 16, 20 | Muscle weight<br>Diameter<br>Axon count<br>Tetanic contraction | Tibialais anterior |
| Vasudevan et al., 2014 | Rat | 24 | Sciatic | 35 | 12 | Muscle weight<br>Axon count | Gastrocnemicus |
| Wakimura et al., 2015 | Rat | 14 (22) | Sciatic | 15 | 24 | Amplitude | |
| Wang, Huang et al., 2014 [56] | Monkey | 20 | Radial | 25 | 20 | NCV | |
| Wang, Itoh et al., 2016 | Rat | 15 | Sciatic | 15 | 24 | Amplitude<br>Diameter | |
| Wang Liu et al., 2010 | Monkey | 12 | Radial | 25 | 20 | Muscle weight<br>NCV | Extensor digitorum longus |
| Wang, Wu et al., 2016 | Rat | 20 (40) | Sciatic | 20 | 4, 8, 12 | SFI<br>Muscle weight<br>Diameter | Gastrocnemicus<br>Triceps surae |
| Wang, Zhao et al., 2012 | Rat | 65 | Sciatic | 15 | 12 | Muscle weight | Triceps surae |
| Whitlock et al., 2009 | Rat | 102 | Sciatic | 14, 28 | 6, 12, 22 | SFI<br>Muscle weight | Gastrocnemicus |
| Xiang et al., 2017 | Rat | 55 | Sciatic | 15 | 12 | SFI<br>Muscle weight<br>NCV | Gastrocnemicus |
| Yan et al., 2016 | Rat | 32 | Sciatic | 20 | 8 | Muscle weight<br>Axon count<br>Tetanic contraction | Extensor digitorum longus |
| Yu, Peng et al., 2009 | Rat | 52 | Sciatic | 10 | 16 | Muscle weight | Triceps surae |
| Yu, Wen et al., 2020 | Rat | 48 | Sciatic | 10 | 6, 12 | SFI<br>NCV<br>Axon count | |
| Zhang, Tong et al., 2008 | Rat | 16 | Sciatic | 10 | 12 | Muscle weight<br>NCV<br>Amplitude | Tibialis anterior |
| Zhang, Zhang et al., 2014 | Rat | 30 | Sciatic | 10 | 2, 4, 6, 8 | SFI<br>Muscle weight<br>NCV<br>Amplitude | Triceps surae |
| Zhao1 et al., 2014 | Rat | 52 | Sciatic | 15 | 12 | Muscle weight | Triceps surae |
| Zhao2 et al., 2011 | Mice | 18 | Sciatic | 10 | 2, 4, 6, 8 | SFI<br>Muscle weight | Triceps surae |

*(Continued)*

**Table 1.** (Continued)

| Reference | Species | Animals (grafts) | Nerve | Graft size (mm) | Time points (weeks) | Outcome measurements | Muscle |
|---|---|---|---|---|---|---|---|
| Zhong et al., 2007 | Dog | 15 | Sciatic | 50 | 24 | Ankle angle<br>NCV | |
| Zhou, Zhang et al., 2015 | Rat | 75 | Sciatic | 10 | 4, 16 | SFI<br>Muscle weight<br>NCV | Gastrocnemicus |
| Zhou, He et al., 2014 | Rat | 72 | Sciatic | 18 | 2, 4, 6, 8, 10, 12 | SFI<br>NCV<br>Diameter<br>Axon count | |
| Zhu et al., 2015 | Rat | 72 | Sciatic | 18 | 4 | Muscle weight<br>Diameter<br>Axon count | Gastrocnemicus |

At the moment, there is only one such method that is FDA approved and available to surgeons. It is produced by AxoGen, Inc., Alachua, Florida. AxoGen develops its human allografts by combining proprietary detergent processing and gamma irradiation which removes cellular remnants while minimizing the microstructural damage. Next to that, chondroitin sulfate proteoglycan is enzymatic removed from the endoneural tube system to advance axon regeneration [19]. We noticed that the studies we included, published before 2008, used a minimal decellularization method like mere freeze-thaw cycling. In conclusion, based on the available data and analysis we did, no clear statement could be made as to which decellularization method is superior.

A few human studies show a resemblance in effect between the acellular allograft and the autograft. This would be a significant development because the allograft has a couple of fundamental advantages as opposed to the autograft. It has an unlimited supply that offers an excellent solution, e.g., plexus surgery after a major trauma. In such trauma, there can be insufficient autograft material to repair the nerve deficits. Therefore, in these cases the allograft offers a solution to restore the damages done. It also avoids potential donor site morbidity such as pain and loss of sensation. Next to that, there is the benefit of a shorter operation time. And finally, the off-the shelf availability of the allograft.

The clinical use of commercially available human acellular nerve allografts (AxoGen) for nerve reconstruction has been reported in several case reports and in a more sizeable multicenter study (RANGER) [14, 74–78]. Unfortunately, the multicenter study lacks the opportunity to say anything about the effectiveness of the allograft, because it did not compare it to an autograft.

**Table 2. Summary of the overall analyses.**

| Outcome measurement | SMD (Hedges g) | 95% convidence interval | $I^2$ | No. of comparisions | No. of studies |
|---|---|---|---|---|---|
| Muscle weight | -2.39 | -1.85 to -2.93 | 86% | 45 | 31 |
| Sciatic function index | -1.59 | -0.44 to -2.73 | 93% | 15 | 14 |
| Tetanic contraction | -0.54 | -0.13 to -0.95 | 62% | 14 | 9 |
| Ankle angle | -0.98 | -0.28 to -1.69 | 74% | 7 | 5 |
| Nerve conduction velocity | -2.20 | -1.66 to -2.75 | 73% | 19 | 18 |
| Amplitude | -0.79 | 0.11 to -1.69 | 83% | 9 | 9 |
| Axon count | -1.40 | -0.78 to -2.01 | 86% | 27 | 18 |
| Diameter | -1.10 | -0.32 to -1.89 | 82% | 13 | 11 |

**Table 3. Subgroup analysis for graft length.**

| Outcome measurement | SMD (Hedges g) | 95% convidence interval | P-value | No. of comparisions |
|---|---|---|---|---|
| Muscle weight | | | | |
| • Short vs. long | -2.13 vs. -4.21 | -1.33 to -2.94 vs. -2.87 to -5.55 | 0.045 | 18 vs. 9 |
| • Medium vs. long | -1.92 vs. -4.21 | -1.09 to -2.75 vs. -2.87 to -5.55 | 0.026 | 18 vs. 9 |
| Nerve conduction velocity | | | | |
| • Short vs. long | -2.36 vs. -1.92 | -1.49 to -3.23 vs. -0.88 to -2.95 | 1 | 8 vs. 6 |
| • Medium vs. long | -2.33 vs. -1.92 | -1.23 to -3.43 vs. -0.88 to -2.95 | 1 | 5 vs. 6 |
| Axon count | | | | |
| Short vs. long | -1.46 vs. -1.21 | -0.54 to -2.38 vs. -0.08 to -2.33 | 1 | 13 vs. 9 |
| Medium vs. long | -1.62 vs. -1.21 | -0.14 to -3.11 vs. -0.08 to -2.33 | 1 | 5 vs. 9 |

Safa et al. [77] and Leckenby et al. [78] both reported data from the RANGER study. Safa et al. conducted an analysis of 365 patients with 624 nerve repairs (AxoGen). They found a meaningful sensible and motor function recovery in 82% of cases. Finally, the authors stated that nerve defects up to 7 cm could achieve a useful recovery after treatment. Leckenby et al.

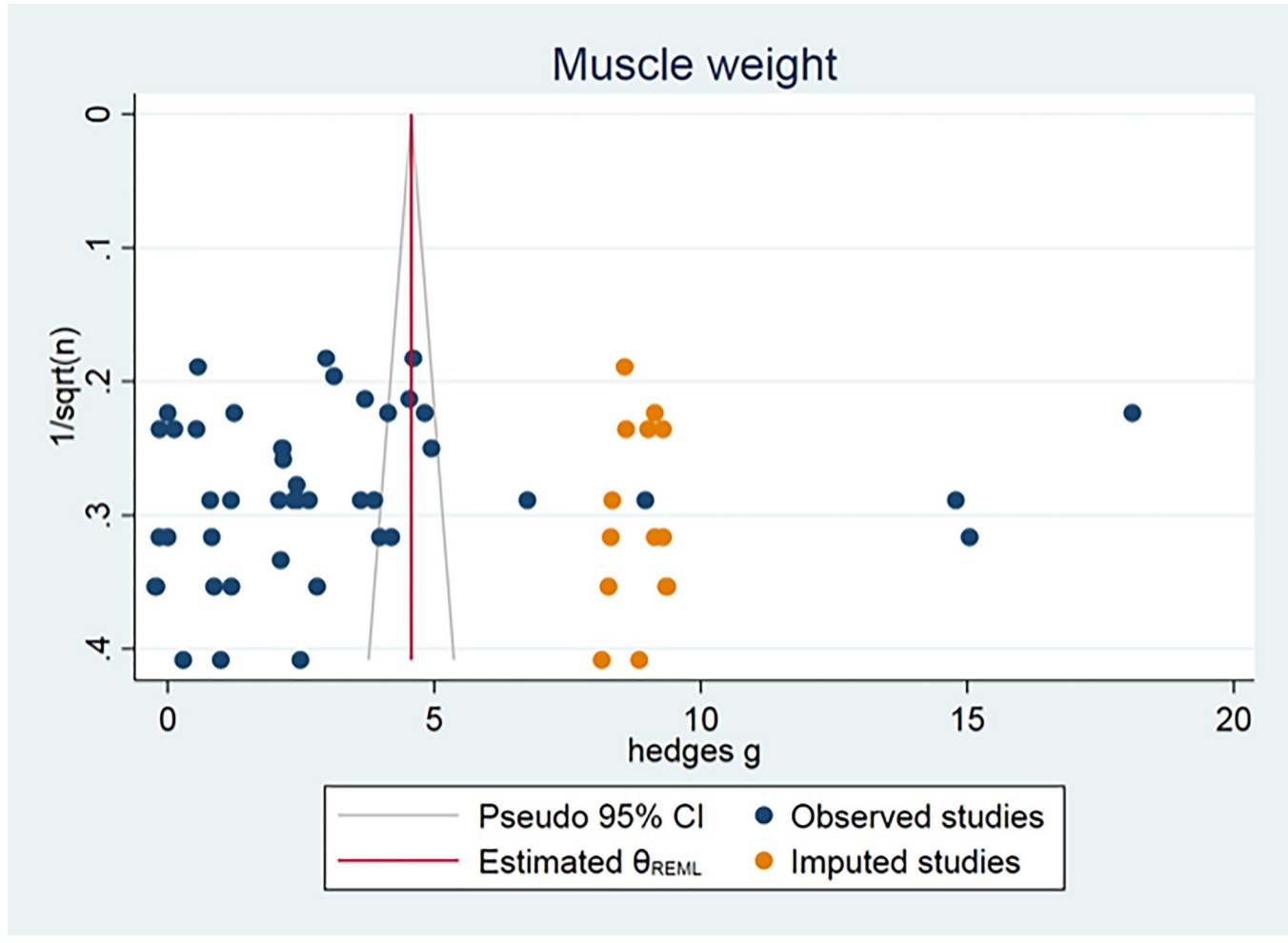

**Fig 4. Muscle weight publication bias.**

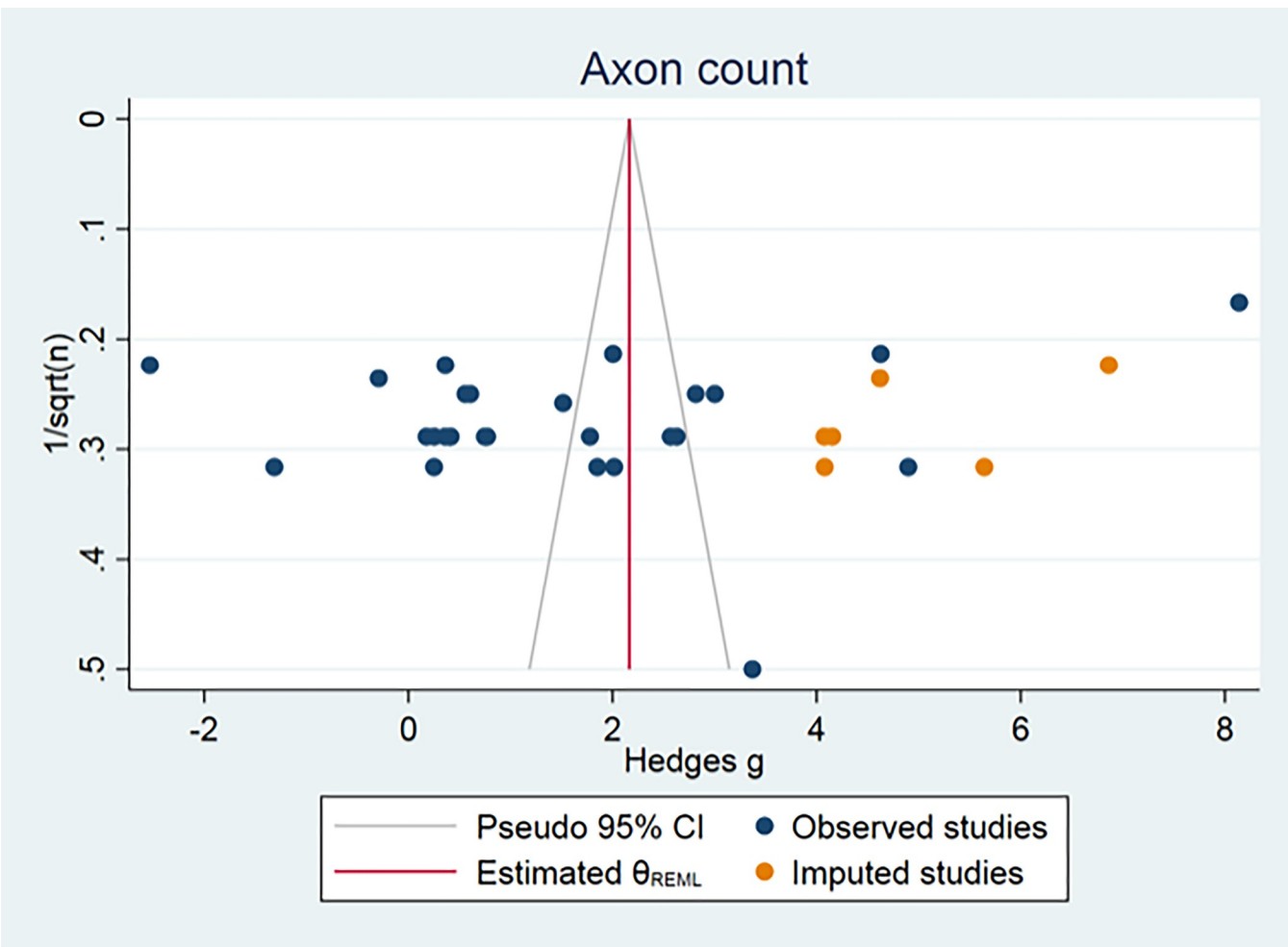

**Fig 5. Axon count publication bias.**

analyzed 171 nerve repairs (AxoGen) in 129 subjects exploring a meaningful sensory and motor function recovery as well. In 73.7% of their cases a meaningful sensory recovery ($^3$S3) was achieved. The percentage of meaningful motor recovery ($^3$M3) was lower at 40,1%, respectively.

Neubauer et al. [19] observed another interesting aspect. They investigated which type of acellular nerve, i.e. motor, sensory or mixed type, is best used for repairing particular types of nerve gaps. No significant differences were found when comparing these three decellularized nerve types as a dominant grafting with regard to axon count and myelinated axons. Most repaired nerves happened to be a sensory dominant one in this study. We noticed that in the animal experiment the opposite is used. The use of motor types was rather common.

Until this day there is no proper "gold standard" to test nerve recovery, although the ultimate goal of nerve recovery is to maximize sensation and motion. The most commonly used outcome measurement for sensation is the von Frey test [79]. For motion, walking track analysis was believed to be the best overall assessment [80–82]. It is rarely used and some would say it is even obsolete. Additionally, walking track analysis does not reflect maximum muscle force capacity. Others say the most precise measurement is the isometric response of muscle to tetanic contraction [83]. The authors are aware that histomorphometry, electrophysiology and

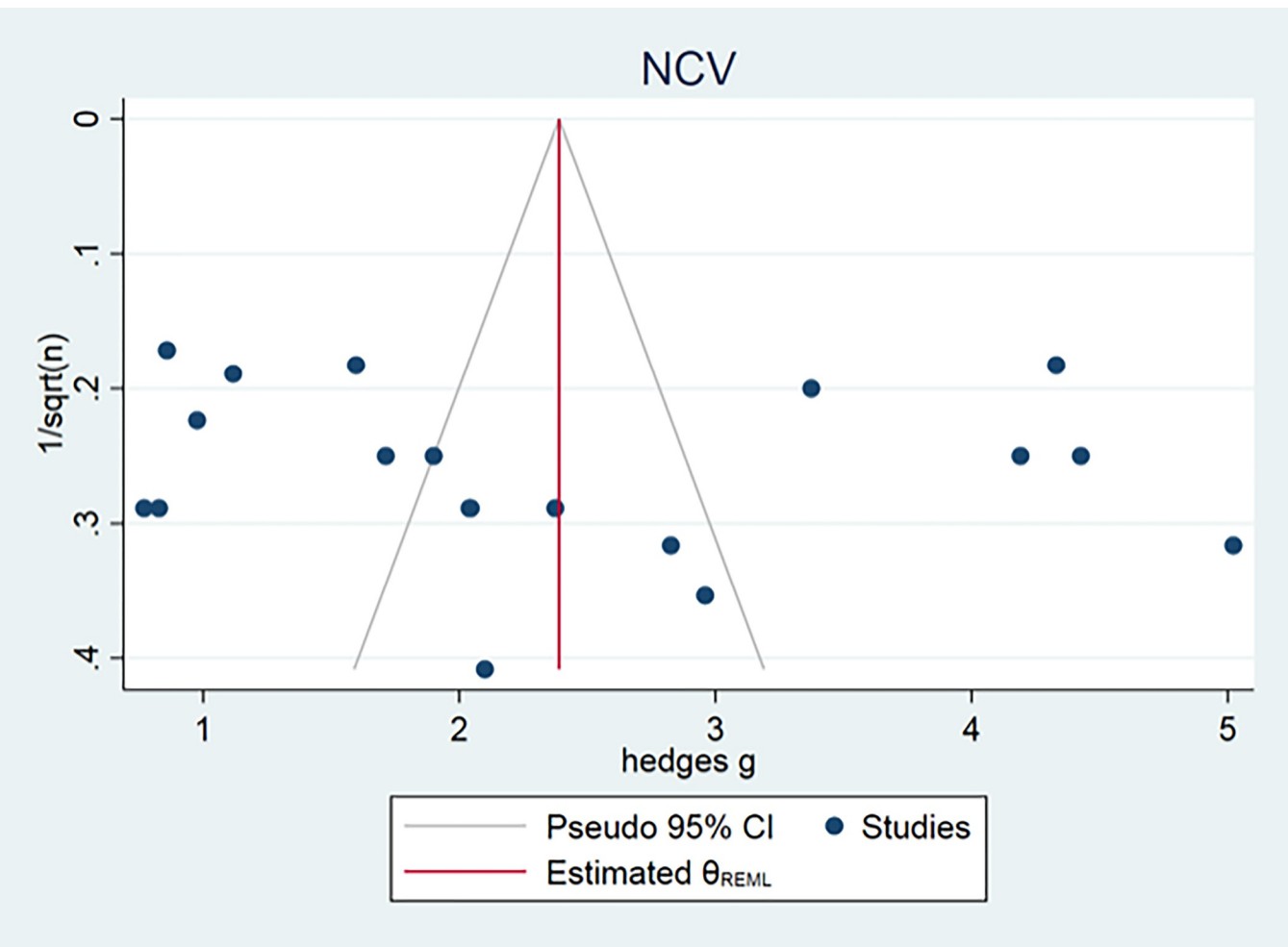

**Fig 6. Nerve conduction velocity publication bias.**

axonal count in particular may have a limited correlation to the real functional recovery of sensation and strength [84]. Next to that, histomorphometry is difficult to compare between different laboratories, because other methods to measure the outcome were used. We used a standardized mean difference for our meta-analysis to compensate for these differences. Over the years methods have evolved from manually calculating axonal count from a light microscopic photograph to a computer calculated estimate. The methods used by the studies in this review vary as well. Searching the publication databases, we found little evidence on which one is the best or on a clear sensitivity or specificity for these methods. However, Kim et al. [85] concluded that the semi-automated method for counting axons in transmission electron microscopic images was strongly correlated with conventional counting methods and showed excellent reproducibility. Nevertheless, the techniques for histomorphometry will always be an estimation and therefore prone to bias.

### Limitations of this review

Firstly, the risk of bias analysis revealed that many essential methodological details were poorly reported in the majority of included studies, which is why most risk of bias items assessed in this analysis were scored as 'unclear risk of bias'. Drawing reliable conclusions from the

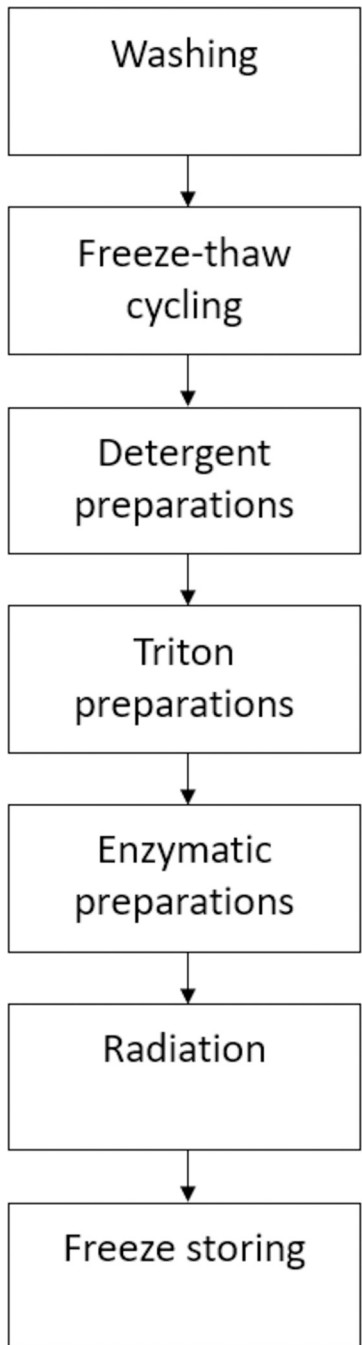

**Fig 7. Global overview decellularization methods.**

included animal studies may have hampered the reliability of the analyses presented in this review.

Secondly, for some outcome measurements the number of included studies in this meta-analysis is relatively low, as a consequence the results of these small meta analyses may be imprecise. Next to that, the heterogeneity between the studies was moderate to high. We used a random effects model, subgroup analyses and conducted two different sensitivity analyses to account for this anticipated heterogeneity.

## Conclusion

This review demonstrates that an acellular nerve allograft results in a significantly inferior nerve recovery compared to an autograft in animal models. In addition, when an allograft is being used an allograft in short and medium (0-1cm, $> 1$-2cm) nerve gaps performs better than an allograft in long ($> 2$cm) nerve gaps. However, the several different animal experiments, investigating the use of acellular allografts, are difficult to compare due to the wide variety of study designs used and the generally poor reporting of essential methodological details. Large population studies or comparative clinical trials with large study populations to allow for participant and injury heterogeneity will be needed to prove and improve the success of the acellular allograft. We strongly advise future animal studies to be designed and reported according to the ARRIVE guidelines [86, 87].

## Supporting information

**S1 Checklist. PRISMA 2020 checklist.**
(PDF)

**S1 Table. Search strategy.**
(DOCX)

**S2 Table. Sensitivity analyses for exclusion of the studies published before 2008.**
(DOCX)

**S3 Table. Sensitivity analysis for exclusion of studies in which animals were their own control.**
(DOCX)

**S1 File. Raw data.**
(XLSX)

## Acknowledgments

The authors would like to thank Mrs On Ying Chan (Health Sciences reference librarian, Radboud University) for assisting with the development of the search strategy.

## Author Contributions

**Conceptualization:** Berend O. Broeren, Caroline A. Hundepool, Ali H. Kumas, Liron S. Duraku, Erik T. Walbeehm, Dominic M. Power, J. Michiel Zuidam, Tim De Jong.

**Data curation:** Berend O. Broeren, Ali H. Kumas.

**Formal analysis:** Berend O. Broeren, Carlijn R. Hooijmans.

**Funding acquisition:** Tim De Jong.

**Investigation:** Berend O. Broeren, Caroline A. Hundepool.

**Methodology:** Berend O. Broeren, Caroline A. Hundepool, Ali H. Kumas, Carlijn R. Hooijmans.

**Project administration:** Berend O. Broeren.

**Supervision:** Tim De Jong.

**Visualization:** Berend O. Broeren.

**Writing – original draft:** Berend O. Broeren.

**Writing – review & editing:** Caroline A. Hundepool, Liron S. Duraku, Erik T. Walbeehm, Carlijn R. Hooijmans, Dominic M. Power, J. Michiel Zuidam, Tim De Jong.

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
