## [Decision Letter · Decision Letter 0]

30 Mar 2023

PONE-D-22-33060The effectiveness of acellular nerve allografts compared to autografts in animal models: a systematic review and meta-analysisPLOS ONE

Dear Dr. Broeren,

Thank you for submitting your manuscript to PLOS ONE. After careful consideration, we feel that it has merit but does not fully meet PLOS ONE’s publication criteria as it currently stands. Therefore, we invite you to submit a revised version of the manuscript that addresses the points raised during the review process.

We look forward to receiving your revised manuscript.

Kind regards,

Panayiotis Maghsoudlou

Academic Editor

PLOS ONE

Journal Requirements:

2. We note that this manuscript is a systematic review or meta-analysis; our author guidelines therefore require that you use PRISMA guidance to help improve reporting quality of this type of study. Please upload copies of the completed PRISMA checklist as Supporting Information with a file name “PRISMA checklist”.

“Conflict of Interest Liron S. Duraku and Dominic M. Power:

Both authors are on the educational committee of the Global Nerve Foundation which is a non-profit organization of which Axogen is a founding Member. Both authors do not get any financial compensation for their role in the Global Nerve Foundation.”

Reviewers' comments:

Reviewer's Responses to Questions

**Comments to the Author**

1. Is the manuscript technically sound, and do the data support the conclusions?

Reviewer #1: Yes

2. Has the statistical analysis been performed appropriately and rigorously? 

Reviewer #1: Yes

3. Have the authors made all data underlying the findings in their manuscript fully available?

Reviewer #1: Yes

4. Is the manuscript presented in an intelligible fashion and written in standard English?

Reviewer #1: Yes

5. Review Comments to the Author

Reviewer #1: The article is a systematic review and meta-analysis of "The efficacy of acellular nerve allografts compared to autografts in animal models".

This is a very interesting topic of study and I believe that the authors have done a great job of searching for information from many papers as demonstrated in the Flow Chart shown in Figure 1 and the literature provided.

I would like to congratulate the researchers for the great work done with this systematic review and meta-analysis as it will serve many other researchers in this field of peripheral nerve regeneration.

I have suggestions for minor changes that I would have to make to the researchers:

1. As this is a systematic review and meta-analysis, I understand that the authors have followed the PRIMA guidelines, but they should specify this in the methodology as they make no mention of these guidelines.

2. I believe that the order of figures 1 and 2 is wrong, as the first time the figures are cited, figure 2 appears first (page 4) and not figure 1 (page 8).

3. Have you considered studying in more detail the decellularizations protocols used for this type of treatment? From your point of view, and after the study carried out, what would you consider to be the best decellularization protocol or technique?

6. PLOS authors have the option to publish the peer review history of their article (what does this mean?). If published, this will include your full peer review and any attached files.

Reviewer #1: No

---

## [Author Response · Author response to Decision Letter 0]

17 Apr 2023

Rebuttal letter

PONE-D-22-33060

“The effectiveness of acellular nerve allografts compared to autografts in animal models: a systematic review and meta-analysis”

Berend O. Broeren1*, Caroline A. Hundepool2, Ali H. Kumas1, Liron S. Duraku3, Erik T. Walbeehm4, Carlijn R. Hooijmans5,6, Dominic M. Power7, J. Michiel Zuidam2, Tim De Jong1 

PLOS ONE

Panayiotis Maghsoudlou

Academic Editor

PLOS ONE

Dear Mr Maghsoudlou,

We thank you and the reviewer for a careful reading and the constructive comments regarding our manuscript and for the opportunity to revise and resubmit. We have addressed all recommendations and suggestions to further improve the manuscript. On behalf of my co-authors, I thank you for considering this revised manuscript for publication. We appreciate your time and look forward to your response.

Yours sincerely,

Berend Broeren (corresponding author)

berend.broeren@xs4all.nl

Journal Requirements:

2. We note that this manuscript is a systematic review or meta-analysis; our author guidelines therefore require that you use PRISMA guidance to help improve reporting quality of this type of study. Please upload copies of the completed PRISMA checklist as Supporting Information with a file name “PRISMA checklist”.

“Conflict of Interest Liron S. Duraku and Dominic M. Power:

Both authors are on the educational committee of the Global Nerve Foundation which is a non-profit organization of which Axogen is a founding Member. Both authors do not get any financial compensation for their role in the Global Nerve Foundation.”

The cover letter was changed and the conflict of interest section was added and extended.

The reference list is complete and correct

Comments to the Author

1. Is the manuscript technically sound, and do the data support the conclusions?

Reviewer #1: Yes

2. Has the statistical analysis been performed appropriately and rigorously? 

Reviewer #1: Yes

3. Have the authors made all data underlying the findings in their manuscript fully available?

Reviewer #1: Yes

4. Is the manuscript presented in an intelligible fashion and written in standard English?

Reviewer #1: Yes

5. Review Comments to the Author

Reviewer #1: The article is a systematic review and meta-analysis of "The efficacy of acellular nerve allografts compared to autografts in animal models".

This is a very interesting topic of study and I believe that the authors have done a great job of searching for information from many papers as demonstrated in the Flow Chart shown in Figure 1 and the literature provided.

I would like to congratulate the researchers for the great work done with this systematic review and meta-analysis as it will serve many other researchers in this field of peripheral nerve regeneration.

We appreciate the positive comments of the reviewer.

I have suggestions for minor changes that I would have to make to the researchers:

1. As this is a systematic review and meta-analysis, I understand that the authors have followed the PRIMA guidelines, but they should specify this in the methodology as they make no mention of these guidelines.

We have added a mention about the fact that the PRISMA guidelines were followed. Page 5 line 108, 109

2. I believe that the order of figures 1 and 2 is wrong, as the first time the figures are cited, figure 2 appears first (page 4) and not figure 1 (page 8).

We have changed the order of figures, figure 2 has become figure 7 on page 14. 

3. Have you considered studying in more detail the decellularizations protocols used for this type of treatment? From your point of view, and after the study carried out, what would you consider to be the best decellularization protocol or technique?

We think this is a fair question and it is one we asked ourselves as well. However, there is little to no evidence for what combination of decellularization methods, give the best nerve regeneration. We tried to investigate which method led to the best nerve recovery by using the data available in the current literature. Due to the great variation in methods used, groups became too small to perform statistical analysis. 

We added information on how the authors think about the question on page 14, 15 line 264-269

---

## [Editor Report · Decision Letter 1]

7 May 2023

The effectiveness of acellular nerve allografts compared to autografts in animal models: a systematic review and meta-analysis

PONE-D-22-33060R1

Dear Dr. Broeren,

We’re pleased to inform you that your manuscript has been judged scientifically suitable for publication and will be formally accepted for publication once it meets all outstanding technical requirements.

Kind regards,

Panayiotis Maghsoudlou

Academic Editor

PLOS ONE

---

## [Editor Report · Acceptance letter]

17 May 2023

PONE-D-22-33060R1 

The effectiveness of acellular nerve allografts compared to autografts in animal models: a systematic review and meta-analysis 

Dear Dr. Broeren:

I'm pleased to inform you that your manuscript has been deemed suitable for publication in PLOS ONE. Congratulations! Your manuscript is now with our production department. 

Kind regards, 

on behalf of

Dr. Panayiotis Maghsoudlou 

Academic Editor

PLOS ONE